# Aqueous proton-selective conduction across two-dimensional graphyne

Le Shi [1], Ao Xu [1], Ding Pan [2,3] & Tianshou Zhao[1]

The development of direct methanol fuel cells is hindered by the issue of methanol crossover across membranes, despite the remarkable features resulting from the use of liquid fuel. Here we investigate the proton-selective conduction behavior across 2D graphyne in an aqueous environment. The aqueous proton conduction mechanism transitions from bare proton penetration to a mixed vehicular and Grotthuss transportation when the side length of triangular graphyne pores increases to 0.95 nm. A further increase in the side length to 1.2 nm results in the formation of a patterned aqueous/vacuum interphase, enabling protons to be conducted through the water wires via Grotthuss mechanism with low energy barriers. More importantly, it is found that 2D graphyne with the side length of less than 1.45 nm can effectively block methanol crossover, suggesting that 2D graphyne with an appropriate pore size is an ideal material to achieve zero-crossover proton-selective membranes.

[1] Department of Mechanical and Aerospace Engineering, HKUST Energy Institute, The Hong Kong University of Science and Technology, Hong Kong, China. [2] Department of Physics and Department of Chemistry, The Hong Kong University of Science and Technology, Hong Kong, China. [3] HKUST Fok Ying Tong Research Institute, Guangzhou, China. Correspondence and requests for materials should be addressed to T.Z. (email: metzhao@ust.hk)

Aqueous proton-selective membranes play a vital role in the development of energy generation and storage systems, such as fuel cells[1] and flow batteries[2]. Conventional polymer-based membranes including Nafion[3], poly-benzimidazole[4] and sulfonated polyether-ether ketones[5] suffer from severe crossover issues, by providing high proton conductivity. The crossover of active species will lower the cycle performance of the energy devices[6,7], and is particularly undesirable for direct methanol fuel cells (DMFC) where methanol crossover can cause cathodic catalyst poisoning and severely deteriorate the cell performance. On the other hand, solid-state proton conductors, such as metal-organic frameworks, can provide high selectivity, but the proton conductivity at room temperature is still far from satisfactory[8]. Thus, a proton-selective membrane that can simultaneously provide high proton conductivity and selectivity is critical for the further development of related energy devices.

Since the first discovery of graphene, two-dimensional (2D) materials have attracted lots of attention from various communities[9,10]. Particularly, 2D materials with artificially created pores become a new choice for gas separation[11] and water desalination[12] membrane. However, creating a large amount of nanoscale pores with narrow size distribution and desired terminations is still a technically formidable task[11,13–15]. In 2014, Geim's group reported that protons can transport across the uniform pores formed by the electron clouds of graphene and h-BN, which opened a new avenue for the design of zero-crossover proton-selective membranes[16]. Follow-up work by our group[17] and Holmes et al.[18] demonstrated reduced crossover of methanol when sandwiching a layer of graphene in Nafion membrane for DMFC. However, whether protons can be conducted through the intrinsic pores formed by the electron clouds of 2D crystals is still controversial[19–23]. The previous work of Achtyl et al.[19] and our previous work[20] show that the energy barriers for aqueous proton conduction across intact graphene or h-BN are too high to be realized at room temperature. The observed proton conductivity in experiment may be mainly attributed to the atomic defects or bias potential.

In addition to graphene and h-BN, there exist many other kinds of 2D materials with intrinsic pores of different geometries, some of which may provide satisfactory proton conductivity and selectivity simultaneously[10]. Graphyne is a family of 2D materials that possess uniform triangular pores with tunable pore size[24,25], providing an ideal platform to study the aqueous proton-selective conduction behavior across porous 2D materials. The structure of graphyne can be regarded as an assembly of phenyl rings connected by acetylenic linkages as illustrated in Fig. 1. The number of acetylenic linkages $n$ can be changed, resulting in different pore sizes. This structure was firstly proposed by Baughman et al.[26] hypothetically in 1987. Afterwards, Li et al.[27] successfully synthesized graphyne ($n = 2$) and graphtetrayne ($n = 4$)[28] in 2010 and 2018, respectively. In this work, graphyne with $n = 1$, 2, 3, and 4 (corresponding to a side length of 0.69, 0.95, 1.20, and 1.45 nm, respectively) were chosen to study the influence of pore size towards the proton-selective conduction behavior. From ab initio molecular dynamics (MD) simulations, we found that the proton-selective conduction behavior in aqueous environment is essentially different from that in vacuum environment. When $n = 1$, a proton in the aqueous phase has to dissociate from a hydronium ion ($H_3O^+$) and form a C–H bond with a carbon atom of graphyne, which corresponds to a high energy barrier of $2.80 \pm 0.03$ eV. When $n = 2$, a proton can penetrate the membrane either in the form of an intact $H_3O^+$ via vehicular mechanism or relay between two water molecules across the membrane via a Grotthuss mechanism, resulting in a relatively lower energy barrier. When $n = 3$ and $n = 4$, water molecules can percolate the pores

of graphyne and form a continuous aqueous phase, where a proton can be conducted via a Grotthuss mechanism, corresponding to low activation energy barriers of $0.27 \pm 0.07$ eV and $0.19 \pm 0.02$ eV, respectively. At the same time, for graphyne ($n = 3$ and $n = 4$), a patterned aqueous/vacuum interphase will be formed, which can effectively block the penetration of other species dissolved in the aqueous phase such as methanol. Based on the calculated energy barriers, graphyne ($n = 4$) can provide a high area-normalized proton conductance as well as an ultrahigh proton/methanol selectivity ($\sim 1.0 \times 10^{12}$).

## Results

**Proton penetration in vacuum environment**. We first calculated the penetration energy barriers of concerned species (bare proton, water, hydronium ion, and methanol) in vacuum environment using climbing image nudged elastic band (CI-NEB) method[29] as benchmark. The calculation results are shown in Fig. 2 and the detailed values are listed in Supplementary Table 1. Graphyne ($n = 1$) shows strong attraction towards bare proton $H^+$, and it exhibits large energy barriers ($> 3.98$ eV) toward all the other species due to its small pore size. For graphyne ($n = 2$), both the attraction towards $H^+$ and the energy barriers toward other species become much smaller. For graphyne ($n = 3$ and $n = 4$), all the species can penetrate graphyne with no energy barrier, and the corresponding energy profiles are flat, indicating that the interaction between graphyne ($n = 3$ and $n = 4$) and the penetration species are weak. Based on these results, for proton transportation in gas phase, graphyne ($n = 1$) is expected to trap $H^+$ in its pores and block the transportation of any other species, while graphyne ($n = 3$ and $n = 4$) cannot provide selectivity toward other species.

**Graphyne in aqueous environment**. Most application scenarios of proton-exchange membranes in energy devices are in aqueous environments at room temperature, where protons mainly exists in the form of $H_3O^+$ and conduct via the Grotthuss mechanism[30,31]. Thus, the interaction between aqueous phase and graphyne determines the possible proton transport mechanism. Here we construct the interphase of water and graphyne by first putting graphyne into a box filled with water with a density of $1\,g\,cm^{-3}$. Then classical MD simulations in $NPT$ ensemble ($P = 1$ atm, $T = 300$ K) were performed for 2 ns with $x$ and $y$ directions fixed. Afterwards, the systems were equilibrized in $NVT$ ensemble ($T = 300$ K) for another 2 ns. The detailed system configuration after the equilibration can be found in Supplementary Table 2. Then, we analyzed the water/graphyne interphase by collecting the data from both a 1 ns MD simulation and a 10 ps ab initio molecular dynamics (AIMD) simulation in $NVT$ ensemble. The water density distribution from the MD and AIMD simulations are shown in Supplementary Figures 3 and 5, which show good agreement with each other. Figure 3 shows the water/graphyne interphases obtained from the 1 ns MD simulation. For graphyne with $n = 1$, a clear wide gap about 4 Å can be observed between the aqueous phase and graphyne. The gap becomes narrower to about 3.5 Å when $n = 2$. For graphyne with $n = 3$ and $n = 4$, water molecules are found to percolate the pores of graphyne and form a patterned aqueous/vacuum interphase. The continuous aqueous phase makes it possible for the protons to be conducted through the water wires across the membrane via Grotthuss mechanism. Although the patterned vacuum phase can help block the penetration of other species dissolved in the aqueous phase such as methanol.

**Aqueous proton conduction across graphyne**. We study the proton conduction behavior across graphyne in aqueous

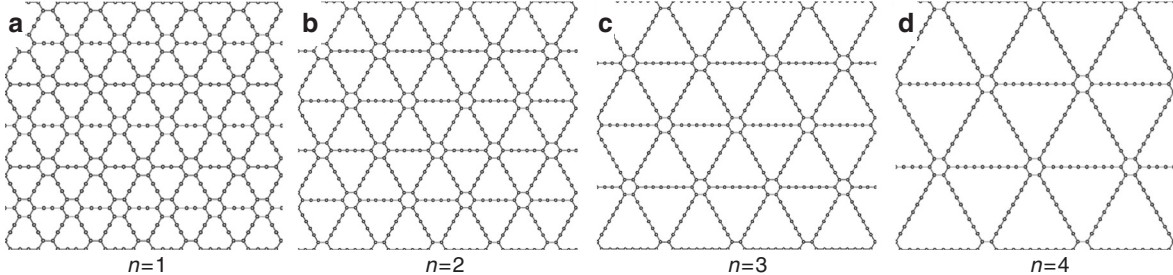

**Fig. 1** Geometries of two-dimensional graphyne. **a** $n = 1$ corresponds to a side length of 0.69 nm. **b** $n = 2$ corresponds to a side length of 0.95 nm. **c** $n = 3$ corresponds to a side length of 1.20 nm. **d** $n = 4$ corresponds to a side length of 1.45 nm

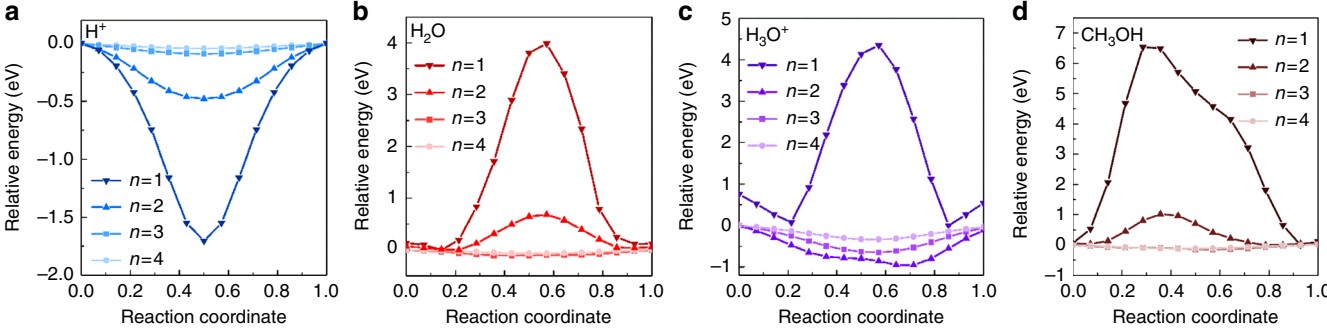

**Fig. 2** Penetration energy barriers of different species across graphyne in a vacuum. **a** proton **b** water **c** hydronium ion **d** methanol

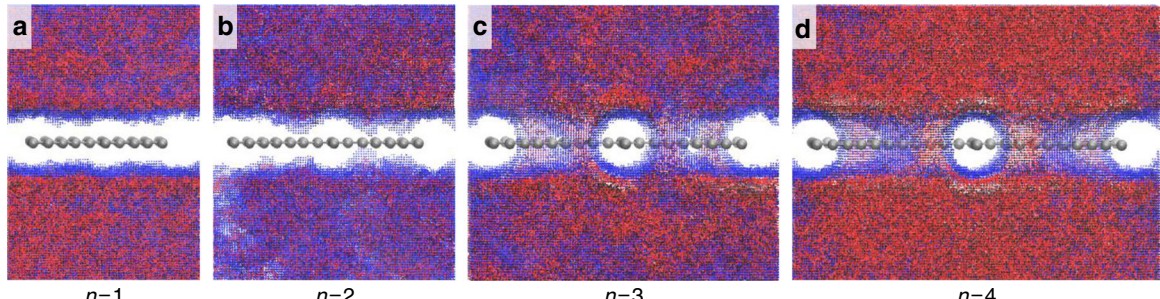

**Fig. 3** Water distribution surrounding graphyne. **a** $n = 1$ **b** $n = 2$ **c** $n = 3$ **d** $n = 4$. The red and blue dots represent the density of oxygen and hydrogen element, respectively

environment using AIMD simulations. An extra proton was put in the aqueous environment and 10 ps unbiased AIMD simulations in *NVT* ensemble were conducted to equilibrize the system. The distance between the proton and graphyne during the unbiased AIMD simulations are plotted in Supplementary Figure 6. As periodical boundary condition was applied, proton can cross the boundary and appear in both sides of graphyne.

We then calculated the proton conduction free energy profiles using ab initio metadynamics[32]. The convergence of deposition pace for each case were tested using the same initial geometry (as listed in Supplementary Tables 5, 7, 9, 11, 13, and 15). Then for each production run, we run three metadynamics simulations using different initial geometries (as listed in Supplementary Tables 6, 8, 10, 12, 14, and 16). For graphyne ($n = 1$ and $n = 2$), as the aqueous phase is not continuous, proton may dissociate from the hydronium ion and become a bare proton or bond with the carbon atoms of graphyne. To track the proton trajectory, we constrained the O–H bonds of all the water molecules when the bond lengths exceed 1.4 nm and adopted the distance between the

extra proton and graphyne as collective variable. A pair of walls were put 4 Å away from graphyne to confine the proton movement as shown in Supplementary Figure 7. The coordination number of the proton with oxygen (O–H) and carbon (C–H) atoms was tracked. Free energy profiles were constructed based on the bias potential deposited as shown in Supplementary Figures 9 and 13. For graphyne ($n = 1$), it is found that every time a proton penetrating graphyne, the O–H coordination number will decrease from one to zero and the C–H coordination number will show a peak exceeds one as shown in Supplementary Figure 8. Based on this result, we propose that for proton penetration across graphyne ($n = 1$), proton will first dissociate from the hydronium ion and attach to the carbon atoms of graphyne, then turn to the other side, leaving the graphyne and further combine with another water molecule to form a hydronium ion, as illustrated in Fig. 4a. The calculated energy barrier for proton penetration across graphyne ($n = 1$) is as high as $2.80 \pm 0.03$ eV as shown in Supplementary Figure 10, which indicates that proton penetration is hard to be realized at room temperature. For graphyne ($n = 2$),

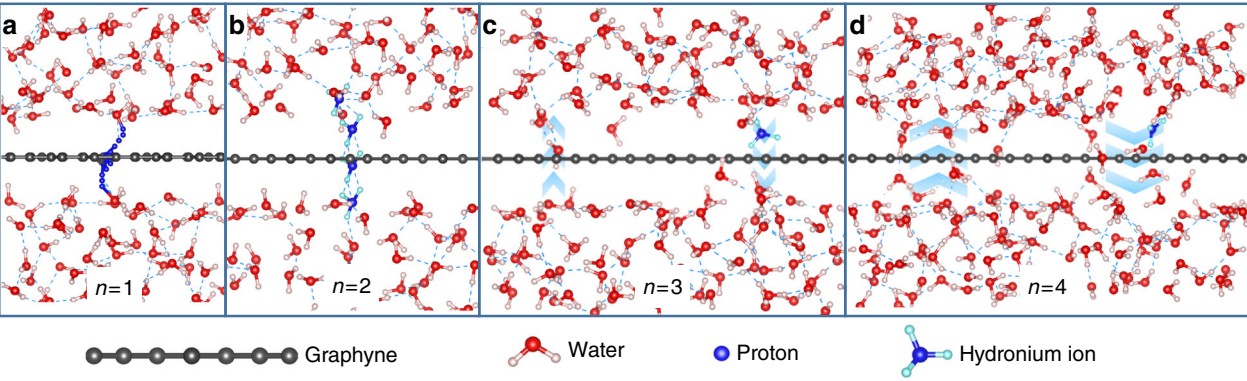

**Fig. 4** Proposed aqueous proton conduction mechanisms across graphyne. **a** $n = 1$ **b** $n = 2$ **c** $n = 3$ **d** $n = 4$

as shown in Supplementary Figure 11, the O–H coordination number oscillates around one all the time, and the C–H coordination number keeps close to zero, which indicates a proton penetration mechanism different from the case of $n = 1$. A further inspection on the index of the oxygen atom connected with proton during the proton penetration process (Supplementary Figure 12) indicates that the proton penetration across graphyne ($n = 2$) can occur via both vehicular and Grotthuss mechanism. As illustrated in Fig. 4b, the proton may either stay attached with a water molecule and penetrate the membrane via vehicular mechanism in the form of hydronium ion or relay between two water molecules via Grotthuss mechanism across the membrane. The calculated energy barrier is $1.30 \pm 0.02$ eV as shown in Supplementary Figure 14. This value is a bit lower than that of graphyne ($n = 1$) but still hard to be achieved at room temperature. As the constraints may result in over-prediction of the proton penetration energy barrier, we also calculated the proton penetration energy barrier across graphyne ($n = 2$) without constraints, as illustrated in Supplementary Figure 15. The distance between the oxygen atom in the hydronium ion and graphyne was chosen as collective variable[33,34], and the free energy profiles were constructed once all the phase space for the collective variable have been visited, as shown in Supplementary Figure 16. The energy barrier was calculated as the difference between the highest energy value when proton crossing the membrane and the mean energy when proton exist in the bulk aqueous phase. The calculated energy barrier for proton penetration is 0.71 eV when all the constraints are removed (as shown in Supplementary Figure 17), smaller than the previous 1.30 eV but still too high to overcome at room temperature.

For graphyne ($n = 3$ and $n = 4$), as the aqueous phase is continuous across the membrane, proton is expected to exist in the form of hydronium ion all the time and be conducted across the membrane mainly via Grotthuss mechanism as illustrated in Fig. 4c, d. The setup for metadynamics simulations are the same with the above case of proton penetration across graphyne ($n = 2$) without constraints. The obtained free energy profiles are shown in Fig. 5. Low energy barriers of $0.27 \pm 0.07$ eV and $0.19 \pm 0.02$ eV were obtained for $n = 3$ and $n = 4$, respectively, which will result in high proton conductance. It should be noted that here we did not consider the nuclear quantum effect of proton, which may result in even lower proton penetration energy barriers[35].

**Methanol penetration across graphyne**. As introduced in the previous section, proton-selective conduction is particularly necessary for DMFC where methanol crossover will lead to catalyst poisoning thus deteriorating the fuel cell system. Therefore, we chose methanol as an example to demonstrate the selectivity

of 2D graphyne-based membranes. A methanol molecule was first put into the aqueous system in place of two water molecules and go through 10 ps unbiased AIMD simulations in $NVT$ ensemble for equilibration. The distance between the carbon atom in the methanol molecule and graphyne during the simulation is shown in Supplementary Figure 20. It can be found that the mobility of methanol is much lower than that of proton. During the 10 ps simulations, the methanol molecule mainly stays around its initial position. Based on the previous calculation results, the proton penetration energy barriers across graphyne ($n = 1$ and $n = 2$) are too high to be achieved at room temperature, thus we did not calculate the free energy profiles for methanol penetration across graphyne when $n = 1$ and $n = 2$. For $n = 3$ and $n = 4$ cases, ab initio metadynamics simulations were performed to calculate the methanol penetration energy barriers. As shown in Supplementary Figure 21, the distance between the carbon atom in the methanol molecule and graphyne was used as collective variable. Similar with the case of proton, the difference between the highest energy when methanol crossing the membrane and the mean energy when methanol exists in the bulk aqueous phase was calculated as energy barrier. The energy profiles are shown in Fig. 6. Energy barriers of $0.82 \pm 0.02$ eV and $0.90 \pm 0.11$ eV were obtained for $n = 3$ and $n = 4$ cases, respectively, which are higher than the proton penetration energy barriers, indicating that the patterned vacuum phase can effectively block the penetration of methanol. The methanol penetration energy barrier for $n = 3$ is a bit smaller than that for $n = 4$. As the length of the triangular pores occupied by aqueous phase in both $n = 3$ and $n = 4$ cases are smaller than the molecular diameter of methanol ($\sim 4.3$ Å), we propose that methanol will squeeze out water molecules in the triangular pores during the methanol penetration process. For the case of $n = 4$, more water molecules will be squeezed out compared with that in the case of $n = 3$, which may result in a higher energy barrier. To validate our hypothesis, we have analyzed the molecular trajectories during the metadynamics simulation. It is found that the methanol molecule will repel water molecules when penetrating graphyne and occupy the entire pore region (as shown in Supplementary Figure 22) in both $n = 3$ and $n = 4$ situations. We also explored the methanol penetration behavior at higher concentrations close to that in real experiment. Six and eight methanol molecules were put into the simulation box to achieve methanol concentrations of 2.22 M and 2.00 M for graphyne ($n = 3$) and graphyne ($n = 4$), respectively. Supplementary Figures 23 and 24 show the trajectory of methanol molecules in unbiased AIMD simulations. It can be found that the behavior of each methanol molecule is similar with the cases of lower concentrations (as shown in Supplementary Figure 20). We then performed ab initio metadynamics simulations on one of the methanol molecules, and the corresponding trajectories are

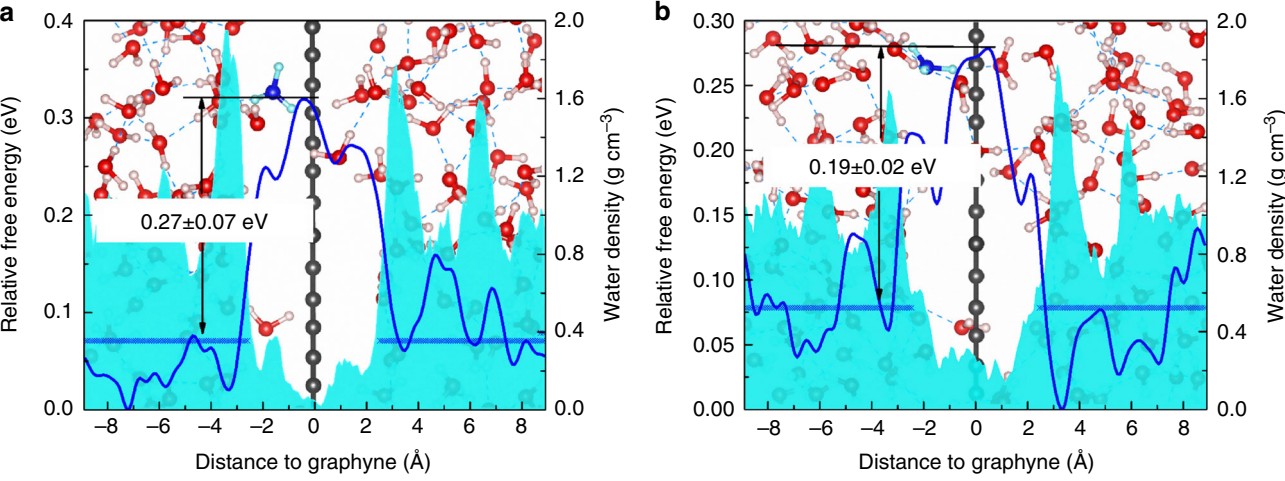

**Fig. 5** Free energy profiles of hydronium ion and density profiles of water. **a** graphyne ($n = 3$) **b** graphyne ($n = 4$). The background shows the initial geometries of the simulation system

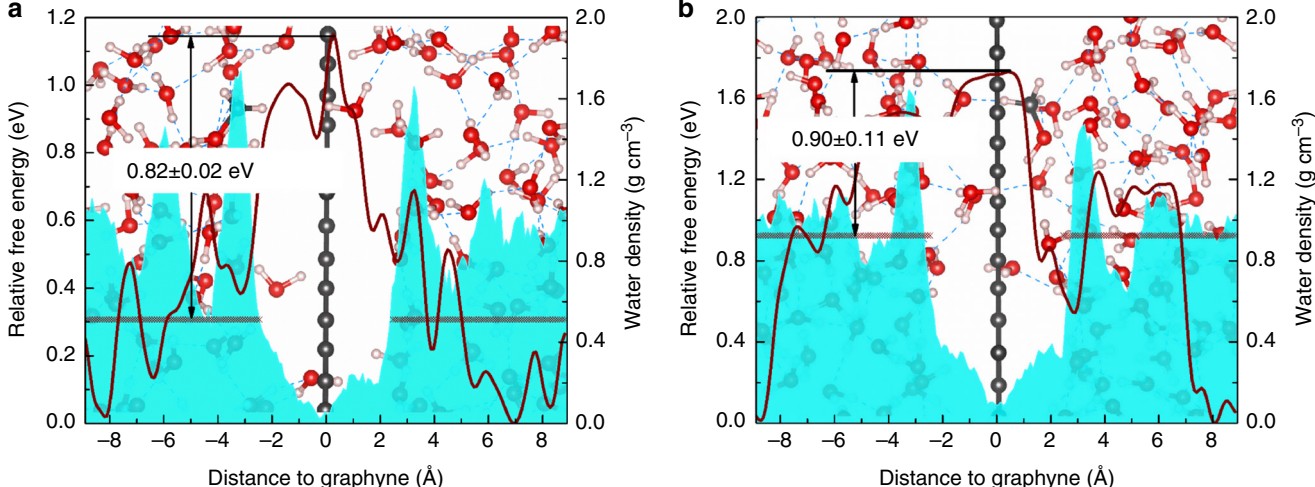

**Fig. 6** Free energy profiles of methanol and density profiles of water. **a** graphyne ($n = 3$) **b** graphyne ($n = 4$). The background shows the initial geometries of the simulation system

shown in Supplementary Figures 25 and 26. It can be found that methanol molecules without bias potential deposited still behave similar with the case of unbiased AIMD simulations. The calculated methanol penetration energy barriers at high concentrations are 1.23 eV and 0.80 eV for graphyne ($n = 3$) and graphyne ($n = 4$), respectively, which are close to the cases of low concentrations.

## Discussion

Based on the above calculation results, graphyne ($n = 3$ and $n = 4$) can provide low energy barriers for proton penetration and high energy barriers for methanol crossover, making them promising candidates for zero-crossover proton-selective membrane. From the calculated energy barriers, we can calculate the area-normalized proton conductance across graphyne ($n = 3$ and $n = 4$) using the Nernst–Einstein[36–41,42] relation:

$$\sigma_{H^+}^* = \frac{F^2}{RT} D_{H^+} C_{H^+} / d \qquad (1)$$

where $F$ is the Faraday constant, $R$ is the ideal gas constant, $T$ is temperature, $d$ is the membrane thickness and $C_{H^+}$ is the concentration of proton. $D_{H^+}$ is the diffusion coefficient of proton,

which can be estimated using the Einstein–Smoluchowski equation[38,40,41,43]:

$$D_{H^+} = \frac{l^2}{\kappa \tau_D} \qquad (2)$$

where $l$ is the mean step distance, and $\kappa$ is a constant depending on the dimensionality of random-walk ($\kappa = 2$, 4, or 6 for one-, two-, and three-dimensional walk). $\tau_D$ is the mean time between successive steps, which can be estimated as[38]:

$$\tau_D = \nu_0^{-1} \exp\left(\frac{\Delta G}{k_B T}\right) \qquad (3)$$

where $k_B$ is the Boltzmann constant, $\nu_0$ is the thermal frequency with $\nu_0 = k_B T / h$, $h$ is the Planck constant, and $\Delta G$ is the effective Gibbs free energy of activation for proton diffusion.

The membrane thickness and the mean step distance for proton penetration are estimated using the length of vacuum phase at the interphase, as shown in Supplementary Figures 27 and 28. The values are 4.85 Å and 4.82 Å for graphyne ($n = 3$) and graphyne ($n = 4$), respectively. Here we take $\kappa = 2$ for the penetration process, thus for proton conduction across graphyne at room temperature

with proton concentration of 1 M, the estimated area-normalized proton conductance for graphyne ($n = 3$) and graphyne ($n = 4$) are $9.66 \times 10^4$ S cm$^{-2}$ and $2.15 \times 10^6$ S cm$^{-2}$, respectively.

An alternative approach to estimate the proton conductance is adopting the ionic conductance model across nanopores, which assumes that ion mobility inside the nanopore is identical to the bulk mobility without restraint[44–46]. For single triangular pore, the conductance can be predicted as[47]:

$$G = \sigma_{\text{bulk}} \left[ \frac{4d}{\sqrt{3}a^2} + \frac{k}{a} \right]^{-1} \qquad (4)$$

where $\sigma_{\text{bulk}}$ is the bulk proton conductivity, $d$ is the membrane thickness, $a$ is the side length of the triangular pore, and $k$ is the geometrical factor ($k = 2.5$ for triangular pores). The area-normalized proton conductance can be estimated as:

$$\sigma^*_{\text{H}^+} = G \cdot \rho \qquad (5)$$

where $\rho$ is the density of pores, which can be calculated from the atomic geometries of graphyne as:

$$\rho = \text{number of pores/area} \qquad (6)$$

The $\rho$ value adopted for $n = 3$ and $n = 4$ graphyne membranes are $2.39 \times 10^{18}$ cm$^{-2}$ and $1.64 \times 10^{18}$ cm$^{-2}$, respectively. We adopted the $\sigma_{\text{bulk}}$ value calculated from the above-mentioned Nernest–Einstein relation using $D_{\text{H}^+} = 9.31 \times 10^{-5}$ cm$^2$ s$^{-1}$[33] and $C_{\text{H}^+} = 1$ M. $d$ is length of vacuum phase at the interphase as obtained before. $a$ was substituted by the mean length of the triangular pores filled with aqueous phase, as shown in Supplementary Figures 27 and 28 and Supplementary Tables 3 and 4. The area-normalized proton conductance estimated using this method are $8.99 \times 10^4$ S cm$^{-2}$ and $3.94 \times 10^5$ S cm$^{-2}$ for graphyne ($n = 3$) and graphyne ($n = 4$), respectively. For both graphyne ($n = 3$) and graphyne ($n = 4$), the area-normalized proton conductance estimated using both methods qualitatively agree with each other, indicating that when the pore size reaches 1.20 nm, the proton conduction behavior inside the pores is similar with that in the bulk phase.

The selectivity of graphyne ($n = 3$ and $n = 4$) is estimated using the Arrehnius equation[11,48] as:

$$S \approx e^{-\frac{\Delta G_{\text{H}^+}}{k_B T}} / e^{-\frac{\Delta G_{\text{CH3OH}}}{k_B T}} \qquad (7)$$

Substituted with the values obtained from our ab initio metadynamics simulations, the selectivity of graphyne ($n = 3$) and graphyne ($n = 4$) are about $2.0 \times 10^9$ and $1.0 \times 10^{12}$, respectively, both of which are high enough to block the methanol crossover. It is worth mentioning that our proposed proton-selective conduction mechanism is not only limited to 2D graphyne materials, but also expected to be suitable for any other inert 2D materials with similar pore sizes, such as the recently synthesized nanoporous graphene[49], 2D-conjugated aromatic polymer[50], and 2D covalent organic thin films[51,52].

In conclusion, we use graphyne as a platform and comprehensively explore the aqueous proton-selective conduction behavior across porous 2D materials. The proton transportation mechanisms change along with the pore sizes. When the pore size exceeds a threshold value (1.20 nm for the triangular pores of graphyne), water molecules can percolate the 2D materials and protons can be conducted via a Grotthuss mechanism, which can provide a high area-normalized proton conductance. At the same time an aqueous/vacuum patterned interphase will be formed, which has been proven to be an effective blockage toward the crossover of other species dissolved in the aqueous phase such as methanol. From our calculation results, graphyne ($n = 4$) can provide high area-normalized proton conductance and an ultra-high selectivity towards methanol, which provides a new

possibility for the design of zero-crossover proton-selective membrane. In practical applications, the porous 2D materials can be sandwiched in porous substrate as reported elsewhere[11,17,18,53], and the atomic defects can be mediated by stacking several layers of nanoporous 2D materials[11].

## Methods

**Density functional theory calculations**. The initial geometries of graphyne were taken from Bartolomei et al.[54]. and then optimized using Abinit[55–57] software package. Perdew–Burke–Ernzerhof (PBE) generalized gradient approximation (GGA)[58] was adopted to describe the exchange correlation functional along with projector-augmented-wave method[59] to describe the electron–ion interactions. Grimme's D3 correction[60] was employed to describe the van der Waals interaction. The cutoff energy was set to be 20 Ha, and the $k$-point mesh was set to be < 0.05 Å$^{-1}$. All the structures were fully optimized to reach a force tolerance of 0.01 eV Å$^{-1}$. Fifteen images were used for the CI-NEB method. For proton penetration, the initial and final images were fixed 3 Å away from the center of the graphyne's pore. For other species, all the images were relaxed.

**MD simulations**. MD and AIMD simulations were carried out using CP2K[61] software package. In the MD simulations, flexible TIP3P water model[62] was used. The carbon atoms of graphyne were fixed to their initial positions and were modeled as uncharged Lennard–Jones (LJ) particles with a cross section $\sigma_{\text{C–C}} = 0.34$ nm and a depth of the potential well of $\varepsilon_{\text{C–C}} = 0.3612$ KJ mol$^{-1}$[63]. The LJ parameters for water-graphyne interaction were determined from the Lorentz–Berthelot mixing rules[64], $\varepsilon_{i,j} = \sqrt{\varepsilon_{i,i}\varepsilon_{j,j}}$ and $\sigma_{i,j} = (\sigma_{i,i} + \sigma_{j,j})/2$. Noše–Hoover chains thermostat[65] with a time constant of 0.1 ps was used to keep the temperature at $T = 300$ K. In AIMD simulations, ab initio Born-Oppenheimer MD was used for the propagation of classical nuclei. The convergence criterion was set to be $1 \times 10^{-7}$ a.u. for the optimization of wave function. Using the Gaussian and plane waves method, the wave function was expanded in the Gaussian double zeta with valence polarization functions basis set. An auxiliary basis set of plane waves was used to expand the electron density up to a cutoff of 400 Ry. The core electrons were treated using PBE gradient correction[58] and Goedecker–Teter–Hutter pseudopotentials[66]. Density functional theory (DFT)-D3[60] correction was used to account for the van der Waals interaction. All the hydrogen atoms were replaced with deuterium and a time step of 0.5 fs was adopted. It is widely reported that DFT calculations with GGA functionals yield over-structured liquid at ambient conditions[67–69], and a temperature around 400 K was necessary for PBE functional to obtain an oxygen–oxygen pair-correlation function comparable with that of experiment at room temperature. We tested the effect of temperature towards the oxygen–oxygen pair-correlation function using a cubic box containing 64 water molecules (as shown in Supplementary Figure 4), the correlation function agrees well with the experimental data when $T = 400$ K. Thus, for the AIMD simulations in $NVT$ ensemble, the Noše–Hoover chains thermostat with a time constant of 0.1 ps was used to keep the temperature at $T = 400$ K. Graphyne was observed to show a slightly curved structure when all the atoms were fully relaxed in the AIMD simulation, making it difficult to identify the occurrence of proton/methanol penetration. Thus, in all the AIMD simulations for production, the carbon atoms in graphyne were fixed.

**Metadynamics simulations**. The metadynamics simulations were performed using CP2K[58] software package with PLUMED[70] plugin. The detailed definition of the collective variables for the metadynamics simulations can be found in Supplementary Note 1. The deposition pace convergence test results were listed in Supplementary Tables 5, 7, 9, 11, 13, and 15. For each case, three metadynamics simulations with different initial geometries were performed as listed in Supplementary Tables 6, 8, 10, 12, 14, and 16. The width, height, and deposition frequency of Gaussian "hills" employed in production runs were listed in Supplementary Table 17.

## Data availability

The data that support the plots within this paper and other finding of this study are available from the corresponding author upon reasonable request.

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

## Acknowledgements

The work described in this paper was fully supported by a grant from the Research Grants Council of the Hong Kong Special Administrative Region, China (Project No. T23-601/17-R). The computations in this work were performed on the Tianhe-2 provided by National Super Computer Center in Guangzhou. D.P. acknowledges support from the Croucher Foundation through the Croucher Innovation Grant.

## Author contributions

T.Z. supervised the research; L.S. conceived the research and designed the numerical simulations; L.S., A.X. and D.P. analyzed the data; L.S. and T.Z. wrote the paper and all the authors discussed the manuscript.

## Additional information

**Competing interests:** The authors declare no competing interests.

