## [Peer Review File · Nature Communications]

Reviewers' comments:

Reviewer #1 (Remarks to the Author):

The authors report a computational study devoted to assess the use of 2-D graphyne carbon sheets as zero-crossover proton-selective membranes for direct methanol fuel cell (DMFC).

The work mostly exploits ab initio molecular dynamics simulations to obtain free energy profiles for the crossing of the hydronium ion and methanol through model 2D membranes with nano-pores of increasing size. It is found that graphynes with the largest pores enable an easy proton conduction as well as avoiding the passage of methanol.

I find the methodology adequate and the paper well written and presented. However, I consider that the main conclusions of the paper are not sufficiently well assessed for two main reasons.

- At first sight, the energy barriers in Fig. 1 for the hydronium ion and methanol and for the largest pores ($n=3,4$) seem not consistent with those reported in Figs. 5 and 6.

It can be expected that the free energy profiles reported in the latter Figures take into account for "collective" contributions that can increase the penetration barrier, but I would also expect that this increase could depend on the solute concentration.

In this regard, the solute concentrations in the simulations are quite smaller than those (2-3 molar) normally employed in DMFC.

I would suggest the authors to perform equivalent simulations with higher solute fractions in order to better assess the possible effect on the free energy penetration profiles.

- I may be wrong but it seems to me that the estimation of the free energy profiles (especially the most relevant ones reported in Figs. 5 and 6) and related activation barriers relies on a single simulation run starting from a well defined initial state. If that is so, then I would say that these results lack of a proper statistical analysis that could significantly alter the reported barrier.

In this case, I would suggest the authors to perform an additional number of simulations starting from different initial states in order to probe possible different behaviours and outcomes.

In my opinion, in order to be considered for publication the manuscript should be revised by taking into account for the above reported comments.

Moreover, the following minor revisions should be also considered:

- line 20. The sentence "The findings presented in the present work.." sounds awkward. The final section of the abstract could be better arranged.

- line 94. "...mainly exist..". It should be "...mainly exists.."

- line 105. "Figure 2 shows..". It should be "Figure 3"

- line 129. "...were tracked." It should be "...was tracked"

- line 131. I would insert "a" after "every time"

- line 147. It reads "1.95 eV" while in Fig. S14 is "1.94 eV"

- line 185. "...methanol exist" should be "...methanol existx"

- line 199. For the proton conductivity estimation a 1M solute concentration is considered while for the effective free energy of activation a value referring to a lower concentration (that of the simulation) is consider. Could this inconsistency affect the reported proton conductivity?

- line 239. It seems to me that in Eq. (5) the minus sign is omitted at the exponents.

- line 252. I would put "At" before "The same time"

- lines 265 and 283. In one case the D2 Grimme correction is said to be used while in the other case (the AIMD simulations) the D3 version is reported. Is there any reason for this?

Reviewer #2 (Remarks to the Author):

The authors report on a computational study of proton transport from aqueous media across graphyne. Graphene is "a theorized allotrope of carbon" (from the Wikipedia page) consisting of phenyl rings linked by polyacetylene units, in a planar two-dimensional structure. Graphyne is similar to graphene insofar as it is a 2D allotrope of carbon. It differs from graphene because it has larger pores, for which size is theoretically adjustable via the length of the polyacetylene linkages. The authors use combinations of several computational approaches to simulate structure and dynamics associated with proton solvation / desolvation at the water / graphyne interfaces, and proton transmission across the interface. They interpret their findings in terms of a proton conductivity of graphyne.

The work is very timely insofar as it addresses recent findings of high proton conductance through graphene and related 2D materials, and potential applications based on this observation. Having said that, the work deals with a material that has never been made and for which stability is only theorized. I think that their paper should say up front that graphyne is a hypothetical material, not a real material on a par with graphene. Readers should not have to look this information up to judge the significance of the work.

The computations themselves seem fine to me, but I think the authors should have a closer look at the interpretation, on pages 11 and 12, in which they use calculational results to obtain proton conductivities for graphyne. By way of illustration, their use of equations 1-3 is problematic. The only things from their calculations that are used in these equations are the activation energy ΔG , and the mean step distance l . Everything else is a constant. These values are not expected to vary more than about a factor of ten from one system to another, yet, they suggest that the variable activation energies they get, from 0.2 to 3.5 eV, should cause proton conductivities to vary by many orders of magnitude for different systems. There is no way you could use equations 1-3 to predict conductivity variations over many orders of magnitude, for the values of activation energy and step length that they are considering. In my opinion, the problem lies in their use of equations intended for calculating bulk conductivities, in units of Siemens / cm, when what they really want are area-normalized conductances, in units of S / cm². Graphyne and graphene are 2D materials for which the meaningful metric is flux, not conductivity. It is not meaningful to discuss a bulk-phase conductivity in such a material. Indeed, it is not meaningful to consider a proton concentration in such a material. Concentration in this context is a property of a bulk phase. For a 2D material one must consider the flux through the material, thinking of it as a dimensionless plane.

The same problem arises in their use of equation 4, which as they note is intended to estimate conductance, in units of siemens, across a single nanopore. A quick dimensional analysis of equation 4 shows that it does indeed give G , conductance, in units of siemens. Yet, they report findings from the equation in units siemens / cm. How did they get values with these units? I don't follow that.

The significance of these computational findings lies in their relevance to experimental findings, and the experimental findings on proton transmission through 2D materials report values in units of area-normalized conductance, S / cm². I think that these authors should interpret their findings in the same way.

Response to the first Reviewer's Comments

The authors report a computational study devoted to assessing the use of 2-D graphyne carbon sheets as zero-crossover proton-selective membranes for direct methanol fuel cell (DMFC). The work mostly exploits *ab initio* molecular dynamics simulations to obtain free energy profiles for the crossing of the hydronium ion and methanol through model 2D membranes with nano-pores of increasing size. It is found that graphynes with the largest pores enable an easy proton conduction as well as avoiding the passage of methanol. I find the methodology adequate and the paper well written and presented. However, I consider that the main conclusions of the paper are not sufficiently well assessed for two main reasons.

Comment#1:

At first sight, the energy barriers in Fig. 1 for the hydronium ion and methanol and for the largest pores ($n=3,4$) seem not consistent with those reported in Figs. 5 and 6. It can be expected that the free energy profiles reported in the latter Figures take into account for "collective" contributions that can increase the penetration barrier, but I would also expect that this increase could depend on the solute concentration. In this regard, the solute concentrations in the simulations are quite smaller than those (2-3 molar) normally employed in DMFC. I would suggest the authors to perform equivalent simulations with higher solute fractions in order to better assess the possible effect on the free energy penetration profiles.

Response:

We thank the reviewer for the suggestion. In the revised manuscript, we tested the convergence of deposition pace for the metadynamics simulations and employed longer deposition pace and smaller Gaussian "hills" in the production run (as listed in Table S17). It is found that the methanol penetration energy barriers across graphyne ($n=3$) and graphyne ($n=4$) are 0.82 ± 0.02 eV and 0.90 ± 0.11 eV, respectively, which are close to each other and smaller than our previously reported values. The updated results indicate that for large pores, the pore size show little influence towards the penetration energy barrier, which is consistent with the NEB calculations in the vacuum environment. We propose that the penetration barrier mainly come from the patterned vacuum/aqueous interphase.

To test the influence of methanol concentration, we further performed *ab initio* molecular dynamics (AIMD) and metadynamics simulations with higher concentration. Six and eight methanol molecules were put into the simulation box to achieve methanol concentrations of 2.22 M and 2.00 M for graphyne ($n=3$) and graphyne ($n=4$), respectively. Figure R1 and R2 show the trajectory of methanol molecules in unbiased AIMD simulations. It can be found that the behaviour of each methanol molecule is similar with the cases of lower concentrations (as shown in Figure S18). We then performed *ab initio* metadynamics simulations on one of the methanol molecules,

and the corresponding trajectories are shown in Figure R3 and R4. It can be found that methanol molecules without bias potential deposited still behave similar with the cases of unbiased AIMD simulations. The calculated methanol penetration energy barriers at high concentrations are 1.23 eV and 0.80 eV for graphyne ($n=3$) and graphyne ($n=4$), respectively, which are close to the cases of low concentrations.

Figure R1 Distance between methanol molecules and graphyne ($n=3$) as a function of time in unbiased *ab initio* molecular dynamics simulations for 10 ps

Figure R2 Distance between methanol molecules and graphyne ($n=4$) as a function of time in unbiased *ab initio* molecular dynamics simulations for 10 ps

Figure R3 Distance between the carbon atoms in methanol molecules and graphyne ($n=3$) as a function of time in the metadynamics simulation of methanol penetration across graphyne ($n=3$).

Figure R4 Distance between the carbon atoms in methanol molecules and graphyne ($n=4$) as a function of time in the metadynamics simulation of methanol penetration across graphyne ($n=4$).

Comment#2:

I may be wrong, but it seems to me that the estimation of the free energy profiles (especially the most relevant ones reported in Figs. 5 and 6) and related activation barriers relies on a single simulation run starting from a well-defined initial state. If that is so, then I would say that these results lack of a proper statistical analysis that could significantly alter the reported barrier.

In this case, I would suggest the authors to perform an additional number of simulations starting from different initial states in order to probe possible different behaviours and outcomes.

Response:

We thank the reviewer for the suggestion. In the revised manuscript, we first tested the convergence of deposition pace for each case using the same initial geometry as listed in Table R1, R3, R5, R7, R9 and R11. For each production run, we run three metadynamics simulations using different initial geometries as listed in Table R2, R4, R6, R8, R10 and R12. The penetration energy barriers are presented in the “average value \pm standard deviation” form in the revised manuscript.

Table R1 Deposition pace convergence test for proton penetration across graphyne ($n=1$)

Pace (fs)	5	10	20
Energy barrier (eV)	3.40	2.83	2.71

Table R2 Calculated proton penetration energy barriers with different initial geometries ($n=1$)

Initial geometry	#1	#2	#3	Average
Energy barrier (eV)	2.83	2.80	2.77	2.80 \pm 0.03

Table R3 Deposition pace convergence test for proton penetration across graphyne ($n=2$)

Pace (fs)	5	10	20
Energy barrier (eV)	1.66	1.29	1.29

Table R4 Calculated proton penetration energy barriers with different initial geometries ($n=2$)

Initial geometry	#1	#2	#3	Average
Energy barrier (eV)	1.29	1.28	1.32	1.30 \pm 0.02

Table R5 Deposition pace convergence test for proton penetration across graphyne ($n=3$)

Pace (fs)	10	20	40
Energy barrier (eV)	0.42	0.25	0.25

Table R6 Calculated proton penetration energy barriers with different initial geometries ($n=3$)

Initial geometry	#1	#2	#3	Average
Energy barrier (eV)	0.25	0.21	0.34	0.27 \pm 0.07

Table R7 Deposition pace convergence test for proton penetration across graphyne ($n=4$)

Pace (fs)	10	20	40
Energy barrier (eV)	0.21	0.17	0.16

Table R8 Calculated proton penetration energy barriers with different initial geometries ($n=4$)

Initial geometry	#1	#2	#3	Average
Energy barrier (eV)	0.17	0.20	0.21	0.19±0.02

Table R9 Deposition pace convergence test for methanol penetration across graphyne ($n=3$)

Pace (fs)	5	10	20
Energy barrier (eV)	1.03	0.80	0.82

Table R10 Calculated methanol penetration energy barriers with different initial geometries ($n=3$)

Initial geometry	#1	#2	#3	Average
Energy barrier (eV)	0.80	0.81	0.84	0.82±0.02

Table R11 Deposition pace convergence test for methanol penetration across graphyne ($n=4$)

Pace (fs)	5	10	20
Energy barrier (eV)	1.14	0.81	0.80

Table R12 Calculated methanol penetration energy barriers with different initial geometries ($n=4$)

Initial geometry	#1	#2	#3	Average
Energy barrier (eV)	0.81	1.02	0.86	0.90±0.11

Comment#3:

Moreover, the following minor revisions should be also considered:

- line 20. The sentence "The findings presented in the present work.." sounds awkward. The final section of the abstract could be better arranged.

- line 94. "..mainly exist..". It should be "..mainly exists.."

- line 105. "Figure 2 shows..". It should be "Figure 3"

- line 129. *"..were tracked." It should be "..was tracked"*

- line 131. *I would insert "a" after "every time"*

- line 147. *It reads "1.95 eV" while in Fig. S14 is "1.94 eV"*

- line 185. *"..methanol exist" should be "..methanol exists"*

- line 199. *For the proton conductivity estimation, a 1M solute concentration is considered while for the effective free energy of activation a value referring to a lower concentration (that of the simulation) is consider. Could this inconsistency affect the reported proton conductivity?*

- line 239. *It seems to me that in Eq. (5) the minus sign is omitted at the exponents.*

- line 252. *I would put "At" before "The same time"*

- lines 265 and 283. *In one case the D2 Grimme correction is said to be used while in the other case (the AIMD simulations) the D3 version is reported. Is there any reason for this?*

Response:

We thank the reviewer for the suggestion. All the grammar errors and typos were corrected in the revised manuscript.

For the proton conductivity, we adopted two methods to calculate the detailed values. For the first method using Nernst-Einstein relation, we used the energy barriers obtained from metadynamics simulations with lower proton concentrations and assumed that the proton penetration energy barriers won't be significantly affected by the proton concentration. Due to the complexity of the collective variable defined to describe the proton position, we cannot test the influence of proton concentration by putting more protons into the system. The calculated area-normalized proton conductance using this method are 1.42×10^5 and 2.16×10^6 S/cm² for graphyne ($n=3$) and graphyne ($n=4$) respectively. In addition, we also employed another method named the ionic conductance model to calculate the area-normalized proton conductance, which assumed that that ion mobility inside the nanopore is identical to the bulk mobility without restraint. The resulted area-normalized proton conductance using the second method only depends on the pore geometries instead of the calculated proton penetration energy barriers. The calculated area-normalized proton conductance using the second method are 8.99×10^4 and 3.94×10^5 S/cm² for graphyne ($n=3$) and graphyne ($n=4$) respectively. The calculated area-normalized proton conductance using two different methods agree with each other qualitatively, which indicate that when the pore size reaches 1.20 nm, the proton conduction behaviour across the membrane is similar with that in the

bulk phase. On the other hand, it also indicates that the assumption in our first method (proton penetration energy barriers won't be significantly influenced by the concentration) is reasonable.

In our previous CI-NEB calculations, the abinit software we employed only provided Grimme's D2 correction. In the revised manuscript, we used the newest version of the software which provided Grimme's D3 correction. The calculated penetration energy barriers in vacuum environment are shown in Figure R5. It can be found that the results using Grimme's D3 correction show little difference with the one using Grimme's D2 correction.

Figure R5 Penetration energy barriers of (a) proton (b) water (c) hydronium ion and (d) methanol across graphyne in vacuum environment using Grimme's D3 correction.

We thank the reviewer for the valuable suggestions, which are helpful for improving the quality of this manuscript.

Response to the Second Reviewer's Comments

The authors report on a computational study of proton transport from aqueous media across graphyne. Graphyne is “a theorized allotrope of carbon” (from the Wikipedia page) consisting of phenyl rings linked by polyacetylene units, in a planar two-dimensional structure. Graphyne is similar to graphene insofar as it is a 2D allotrope of carbon. It differs from graphene because it has larger pores, for which size is theoretically adjustable via the length of the polyacetylene linkages. The authors use combinations of several computational approaches to simulate structure and dynamics associated with proton solvation/desolvation at the water/graphyne interfaces, and proton transmission across the interface. They interpret their findings in terms of a proton conductivity of graphyne.

Comment#1:

The work is very timely insofar as it addresses recent findings of high proton conductance through graphene and related 2D materials, and potential applications based on this observation. Having said that, the work deals with a material that has never been made and for which stability is only theorized. I think that their paper should say up front that graphyne is a hypothetical material, not a real material on a par with graphene. Readers should not have to look this information up to judge the significance of the work.

Response:

We thank the reviewer for the suggestion. The structure of graphyne was firstly proposed by Baughman *et al* hypothetically in 1987. [R1] Afterwards, numerous efforts have been devoted into the experimental synthesis of this family of material [R2-R4]. Notably, in 2010, Li *et al* successfully synthesized large area graphyne ($n=2$) films with 3.61 cm^2 on the surface of copper *via* a cross-coupling reaction using hexaethynylbenzene. [R5] Moreover, in 2018, the same group successfully synthesized graphyne ($n=4$) *via* Sonagashira cross-coupling reaction of hexaethynylbenzene and diisobutadiyne [R6]. Thus, it is unfair to say graphyne is still a hypothetical material. In the revised manuscript, corresponding descriptions have been revised and references were added.

Comment#2:

The computations themselves seem fine to me, but I think the authors should have a closer look at the interpretation, on pages 11 and 12, in which they use calculational results to obtain proton conductivities for graphyne. By way of illustration, their use of equations 1-3 is problematic. The only things from their calculations that are used in these equations are the activation energy ΔG , and the mean step distance l . Everything else is a constant. These values are not expected to vary more than about a factor of ten from one system to another, yet, they suggest that the variable activation energies they get, from 0.2 to 3.5 eV, should cause proton conductivities to vary by many orders of magnitude for different systems. There is no way you could use equations 1-3 to predict conductivity variations over many orders of magnitude, for the values of activation energy and step length that they are considering. In my opinion, the problem lies in their use of equations intended for calculating bulk conductivities, in units of Siemens / cm, when what they really want are area-normalized conductances, in units of S / cm^2 . Graphyne and graphene are 2D materials for which the meaningful metric is flux, not conductivity. It is not meaningful to discuss a bulk-phase conductivity in such a material. Indeed, it is not meaningful to consider a proton concentration in such a material. Concentration in this context is a property of a bulk phase. For a 2D material one must consider the flux through the material, thinking of it as a dimensionless plane.

The same problem arises in their use of equation 4, which as they note is intended to estimate conductance, in units of siemens, across a single nanopore. A quick dimensional analysis of equation 4 shows that it does indeed give G , conductance, in units of siemens. Yet, they report findings from the equation in units siemens / cm. How did they get values with these units? I don't follow that.

The significance of these computational findings lies in their relevance to experimental findings, and the experimental findings on proton transmission through 2D materials report values in units of area-normalized conductance, S/cm². I think that these authors should interpret their findings in the same way.

Response:

We thank the reviewer for this suggestion. In our previous manuscript, Equations 1-3 were only used to estimate the proton conductivity across graphyne ($n=3$) and graphyne ($n=4$). The corresponding activation energies were 0.42 eV and 0.21 eV instead of 0.2 to 3.5 eV, and the resulted conductivity did not vary orders of magnitude. For graphyne ($n=1$) and graphyne ($n=2$), as the proton penetration energy barriers are too high to be achieved at room temperature, we did not calculate their proton conductivity.

In the revised manuscript, we run more metadynamics simulations to ensure the convergence of our results. The updated proton penetration energy barrier for graphyne ($n=3$) and graphyne ($n=4$) are 0.27 ± 0.07 and 0.19 ± 0.02 eV respectively. As suggested by the Reviewer, we calculated the area-normalized conductance in units of S/cm² for graphyne ($n=3$) and graphyne ($n=4$) using the two approaches we previously adopted. The calculated area-normalized conductance based on Nernst-Einstein relation are 9.66×10^4 and 2.16×10^6 S/cm² for graphyne ($n=3$) and graphyne ($n=4$), respectively. The calculated area-normalized conductance based on the ionic conductance model across nanopores are 8.99×10^4 and 3.94×10^5 S/cm², respectively. For both graphyne ($n=3$) and graphyne ($n=4$), the area-normalized proton conductance estimated using both methods qualitatively agree with each other, indicating that when the pore size reaches 1.20 nm, the proton conduction behavior across the membrane is similar with that in the bulk phase. The calculation details can be found in the revised manuscript.

We thank the reviewer for the valuable suggestions, which are helpful for improving the quality of this manuscript.

References

- [R1] Baughman, R. H., Eckhardt, H. & Kertesz, M. Structure-property predictions for new planar forms of carbon: layered phases containing sp² and sp atoms. *J. Chem. Phys.* **87**, 6687-6699 (1987).
- [R2] Huang, C., Li, Y., Wang, N., Xue, Y., Zuo, Z., Liu, H. & Li, Y. Progress in research into 2D graphdiyne-based materials. *Chem. Rev.* (2018). DOI: 10.1021/acs.chemrev.8b00288
- [R3] Li, Y., Xu, L., Liu, H. & Li, Y. Graphdiyne and graphyne: from theoretical predictions to practical construction. *Chem. Soc. Rev.* **43**, 2572-2586 (2014).

[R4] Kang, J., Wei, Z. & Li, J. Graphyne and its family: recent theoretical advances. *ACS Appl. Mater. Interfaces* (2018) DOI: 10.1021/acsami.8b03338

[R5] Li, G., Li, Y., Liu, H., Guo, Y., Li, Y. & Zhu, D. Architecture of graphdiyne nanoscale films. *Chem. Commun.* **46**, 3256-3258 (2010).

[R6] Gao, J., Li, J., Chen, Y., Zuo, Z., Li, Y., Liu, H. & Li, Y. Architecture and properties of a novel two-dimensional carbon material-graphtetrayne. *Nano Energy* **43**, 192-199 (2018).

Reviewers' comments:

Reviewer #1 (Remarks to the Author):

The authors addressed all the issues indicated in the first review and provided further calculations and explanations to better support the conclusions of the paper.

The new free energy barriers for methanol penetration (see Fig. 6) are now considerably lower than those previously but still sufficiently larger with respect to those for the hydronium ion to guarantee a high selectivity, as shown from the resulting values of eq. (6).

I consider the paper suitable to be publishable after considering the following minor issues:

- I would expect (considering pore size assumptions) the free energy barrier for methanol penetration for $n=3$ to be larger than that for $n=4$ but this is not the case. A comment on this result would be appreciated.

-On page 10 it is stated that "...are much higher than the proton penetration barrier...". I would put it in lighter way since the energy difference is not dramatic, being around a factor 4.

Reviewer #2 (Remarks to the Author):

The authors present a revised version of their paper on proton and small-molecule transport through graphyne embedded in Nafion. They have clarified several things about the calculations which is helpful. They have changed the discussion on proton conductivity estimation to estimate the area-normalized conductance instead of simple conductivity. This is helpful for making comparisons with other 2D materials. I note that they took equation 3 from reference 43, but in reference 43 it appears this way;

$$t_D = v_0^{-1} \exp [\Delta G / kT]$$

The version in the present manuscript has left off the exponential. I think this must have been a typographical error on the authors' part, leaving off the exponential. When the exponential is included the equation makes more sense. It then predicts conductances different by 20X for $n=3$ and $n=4$, when the activation free energies differ by less than a factor of two. This would not have been possible using the equation as given in the present manuscript.

I'm still not convinced that this analysis for proton conductance is particularly useful. The analysis in Ref 43 was for predicting conductivity (or areal conductance with an assumption about membrane thickness) in Nafion. The particular version they have used here is one that assumes proton conduction by surface diffusion. Activation energies were determined by a summation of terms considering only coulombic effects involving proton and sulfonate groups. Activation energies were fairly low, so, conductance was high. The present authors have simply taken the activation energies from their computations and used the analysis from Ref 43 to predict conductivities. It happens that they also obtained low activation energies, so, they also obtained high conductivities. This is not a great surprise. In the end, regarding proton conduction, what they have done is to show that, for graphyne with $n = 3$ and 4, the graphyne is a poor barrier to protons. I accept that finding and think their calculations do support it.

The key finding of the paper remains the prediction that permeation of methanol is much slower (has a higher activation energy) than protons. The present version makes this finding clear in the text on page 13, and through equation 6. I am a little surprised that the activation energy for methanol transmission across graphyne is as high as it is; I would have guessed that if proton transmission was similar to bulk, that methanol transmission would be too, and there would be little gain in relative

transport rates. But, that is just my instinct; if their calculation says otherwise, I accept that. All that will be left then will be to test by experiment.

One small point, the caption in the new Fig 2 is wrong; a through d refer to the permeant species not the graphyne type.

Reviewer #3 (Remarks to the Author):

Recommendation

The paper is publishable subject to major revisions noted.

In this work, ab initio molecular dynamics simulations have been performed to investigate the performance of 2D γ -graphyne ($n=2,3,4$) sheets as proton-selective conduction membranes for use in direct methanol fuel cells (DMFC). The free energy profiles obtained from ab initio metadynamics simulations suggest that γ -graphynes with larger pores ($n=3,4$) enable proton transfer via the Grotthuss mechanism across the membrane with low energy barriers while blocking the passage of methanol. The simulation results are then employed to compute the proton conductivity and selectivity of the membranes. However, I have the following concerns regarding the methodology and interpretation of the results.

Concerns

1. If I understand correctly, for the calculation of the free energy profile of aqueous proton transfer across the γ -graphyne ($n=2$), only a single hydrogen on the hydronium ion is unconstrained and able to transport across the membrane via Grotthuss mechanism. Since the Grotthuss proton transfer across the membrane involves occurrence of favorable relative orientation between the hydronium ion and water molecule across the membrane, allowing only one of the hydrogens on the hydronium ion to transport across can lead to over-prediction of the free-energy barrier. I would suggest the authors to check if the barrier changes when all of the hydrogen atoms on the hydronium is allowed to transport across the membrane.

Also, the water molecule receiving the extra proton across the membrane is likely to transfer one of its hydrogens to the bulk aqueous phase in a relay fashion, since the extra proton is more stable in the bulk solvated phase rather than on the water molecule at the edge of the vacuum/aqueous interphase. Since this second transfer is constrained to not occur, the free energy profile for the $n=2$ case might not be accurate.

2. In the first method for determining area-normalized proton conductance (equations 1-3), the pore density of the membrane is not being taken into account. Protons are conducted only through the triangular pores, while when you are area-normalizing you have to account for the whole area of the membrane. So, the conductance obtained by equation (1) needs to have a multiplicative factor (number of pores*effective area of pore / area of membrane).

If I understand correctly, this is accounted for in the second method through equation (5) which has ρ (density of pores) as a multiplicative factor.

3. Please provide a discussion on how ρ (density of pores) in equation (5) is computed. Also, provide its value for $n=3$ and $n=4$ graphyne membranes.

The conductance G in equation (4) is in units of Siemens. Given that, it is not clear how area

normalized conductance in equation (5) = $G \cdot \rho$ has units of S/cm².

4. Paper needs editing for English style and usage. For instance,

a. In the abstract line 10, "Despite of its remarkable features resulted from the use of liquid fuel" sounds wrong. It should be 'resulting from' and 'Despite its...'

b. On page 2 line 35, "Along with the emergence of two-dimensional (2D) materials since the first discovery of graphene [9-11], 2D materials with artificially created pores become a new choice of membrane.."

c. On page 3 line 51, "Graphyne is a family of 2D materials possess uniform triangular pores with controllable pore size..."

d. On page 11, line 212, it is 'Nernst-Einstein' and not 'Nernest-Einstein'.

I am only listing a few instances. Please have the paper proof-read for English usage.

Response to the First Reviewer's Comments

The authors addressed all the issues indicated in the first review and provided further calculations and explanations to better support the conclusions of the paper. The new free energy barriers for methanol penetration (see Fig. 6) are now considerably lower than those previously but still sufficiently larger with respect to those for the hydronium ion to guarantee a high selectivity, as shown from the resulting values of eq. (6). I consider the paper suitable to be publishable after considering the following minor issues:

Comment#1:

I would expect (considering pore size assumptions) the free energy barrier for methanol penetration for $n=3$ to be larger than that for $n=4$ but this is not the case. A comment on this result would be appreciated.

Response:

We thank the reviewer for this suggestion. The length of the triangular pores occupied by aqueous phase in both $n=3$ and $n=4$ cases are smaller than the molecular diameter of methanol (~ 4.3 Å). During the methanol penetration process, we propose that methanol will squeeze out water molecules in the triangular pores. For the case of $n=4$, more water molecules will be squeezed out compared with that in the case of $n=3$, which may result in a higher energy barrier. To validate our hypothesis, we have analyzed the molecular trajectories during the metadynamics simulations. It is found that the methanol molecule will repel water molecules when penetrating graphyne and will occupy the entire pore region (as shown in Figure R1, also Figure S22 in the revised manuscript) in both $n=3$ and $n=4$ situations.

Figure R1. Snapshot of methanol penetration across graphyne (a) $n=3$ and (b) $n=4$

Comment#2:

On page 10 it is stated that "...are much higher than the proton penetration barrier...". I would put it in lighter way since the energy difference is not dramatic, being around a factor 4.

Response:

We thank the reviewer for this suggestion. In the revised manuscript, we have rewritten the description as "...are higher than the proton penetration barrier...".

We thank the reviewer for the valuable suggestions, which are helpful for improving the quality of this manuscript.

Detailed Response to the Second Reviewer's Comments

The authors present a revised version of their paper on proton and small-molecule transport through graphyne embedded in Nafion. They have clarified several things about the calculations which is helpful. They have changed the discussion on proton conductivity estimation to estimate the area-normalized conductance instead of simple conductivity. This is helpful for making comparisons with other 2D materials.

Comment#1:

I note that they took equation 3 from reference 43, but in reference 43 it appears this way: $tD = v_0 \exp[-\Delta G/kT]$. The version in the present manuscript has left off the exponential. I think this must have been a typographical error on the authors' part, leaving off the exponential. When the exponential is included the equation makes more sense. It then predicts conductances different by 20X for $n=3$ and $n=4$, when the activation free energies differ by less than a factor of two. This would not have been possible using the equation as given in the present manuscript.

Response:

We thank the reviewer for this comment. This is a typographical error and the exponential symbol has been included in the revised manuscript.

Comment#2:

I'm still not convinced that this analysis for proton conductance is particularly useful. The analysis in Ref 43 was for predicting conductivity (or areal conductance with an assumption about membrane thickness) in Nafion. The particular version they have used here is one that assumes proton conduction by surface diffusion. Activation energies were determined by a summation of terms considering only coulombic effects involving proton and sulfonate groups. Activation energies were fairly low, so, conductance was high. The present authors have simply taken the activation energies from their computations and used the analysis from Ref 43 to predict conductivities. It happens that they also obtained low activation energies, so, they also obtained high conductivities. This is not a great surprise. In the end, regarding proton conduction, what

they have done is to show that, for graphyne with $n = 3$ and 4 , the graphyne is a poor barrier to protons. I accept that finding and think their calculations do support it.

Response:

The Nernst-Einstein equation and Einstein-Smoluchowski equation we adopted in the analysis have a wide range of applications and they are not limited to certain diffusion mechanism. For example, the Nernst-Einstein equation has been adopted to estimate the lithium ion diffusivity in aqueous electrolytes [R1], and both the Nernst-Einstein equation and Einstein-Smoluchowski equation have been adopted to estimate the charge transport in ionic liquids [R2, R3]. New references (Ref 44-46 in revised manuscript) have been added in the revised manuscript.

Comment#3:

The key finding of the paper remains the prediction that permeation of methanol is much slower (has a higher activation energy) than protons. The present version makes this finding clear in the text on page 13, and through equation 6. I am a little surprised that the activation energy for methanol transmission across graphyne is as high as it is; I would have guessed that if proton transmission was similar to bulk, that methanol transmission would be too, and there would be little gain in relative transport rates. But, that is just my instinct; if their calculation says otherwise, I accept that. All that will be left then will be to test by experiment.

Response:

The penetration mechanisms for proton and methanol crossing graphyne are quite different. For protons, they can be conducted *via* the hydrogen bond network across the graphyne, which is the same as that in bulk phase; for methanol, however, as the pore sizes occupied by the aqueous phase are so small, it has to break some hydrogen bonds formed between water molecules and methanol to penetrate, which will result in a higher energy barrier. In addition, during the penetration process, it will squeeze out water molecules in the aqueous pores (as shown in Figure R1, also Figure S22 in revised manuscript), which will make the energy barriers even higher.

Comment#4:

One small point, the caption in the new Fig 2 is wrong; a through d refer to the permeant species not the graphyne type.

We thank the reviewer for this comment. The caption has been corrected in the revised manuscript.

We thank the reviewer for the valuable suggestions, which are helpful for improving the quality of this manuscript.

Detailed Response to the Third Reviewer's Comments

The paper is publishable subject to major revisions noted. In this work, *ab initio* molecular dynamics simulations have been performed to investigate the performance of 2D γ -graphyne

($n=2,3,4$) sheets as proton-selective conduction membranes for use in direct methanol fuel cells (DMFC). The free energy profiles obtained from *ab initio* metadynamics simulations suggest that γ -graphynes with larger pores ($n=3,4$) enable proton transfer via the Grotthuss mechanism across the membrane with low energy barriers while blocking the passage of methanol. The simulation results are then employed to compute the proton conductivity and selectivity of the membranes. However, I have the following concerns regarding the methodology and interpretation of the results.

Comment#1:

If I understand correctly, for the calculation of the free energy profile of aqueous proton transfer across the γ -graphyne ($n=2$), only a single hydrogen on the hydronium ion is unconstrained and able to transport across the membrane via Grotthuss mechanism. Since the Grotthuss proton transfer across the membrane involves occurrence of favorable relative orientation between the hydronium ion and water molecule across the membrane, allowing only one of the hydrogens on the hydronium ion to transport across can lead to over-prediction of the free-energy barrier. I would suggest the authors to check if the barrier changes when all of the hydrogen atoms on the hydronium is allowed to transport across the membrane.

Also, the water molecule receiving the extra proton across the membrane is likely to transfer one of its hydrogens to the bulk aqueous phase in a relay fashion, since the extra proton is more stable in the bulk solvated phase rather than on the water molecule at the edge of the vacuum/aqueous interphase. Since this second transfer is constrained to not occur, the free energy profile for the $n=2$ case might not be accurate.

Response:

we thank the reviewer for this suggestion. In the revised manuscript, we have re-calculated the proton penetration energy barrier across graphyne ($n=2$) without constraints. The parameters adopted for the metadynamics simulation are the same with that of proton penetration across graphyne ($n=3,4$) The distance between hydronium ion and graphyne ($n=2$) can be found in Figure R2 (Figure S16 in revised manuscript) and the corresponding energy profile can be found in Figure R3 (Figure S17 in revised manuscript). The calculated energy barrier for proton penetration is 0.71 eV when all the constraints are removed, which is smaller than previous value of 1.30 eV. However, the energy barrier is still too high to overcome at room temperature.

Figure R2. Distance between hydronium ion and graphyne ($n=2$) (blue marks) and the coverage of visited hydronium positions in the phase space (green line) as a function of time in the metadynamics simulations of proton penetration across graphyne ($n=2$)

Figure R3. Free energy profile of hydronium ion and density profile of water as a function of distance to graphyne ($n=2$). The background shows the initial geometry of the simulation system.

Comment#2:

In the first method for determining area-normalized proton conductance (equations 1-3), the pore density of the membrane is not being taken into account. Protons are conducted only through the triangular pores, while when you are area-normalizing you have to account for the whole area of

*the membrane. So, the conductance obtained by equation (1) needs to have a multiplicative factor (number of pores*effective area of pore / area of membrane).*

If I understand correctly, this is accounted for in the second method through equation (5) which has ρ (density of pores) as a multiplicative factor.

Response:

In the first method for determining area-normalized proton conductance, we used the proton penetration energy barriers calculated from the metadynamics simulations. The influence of pore ratios was already included in the calculation of these energy barriers, where we did not separate the pore and non-pore area. Thus, in the following estimation process, the multiplicative factor of pore ratio is not needed.

Comment#3:

Please provide a discussion on how ρ (density of pores) in equation (5) is computed. Also, provide its value for $n=3$ and $n=4$ graphyne membranes. The conductance G in equation (4) is in units of Siemens. Given that, it is not clear how area normalized conductance in equation (5) = $G \cdot \rho$ has units of S/cm².

Response:

We thank the reviewer for this suggestion. The ρ (density of pores) was computed by:

$$\rho = \text{number of pores/area}$$

As ρ calculated in this way has a unit of number/cm², the area normalized conductance in equation (5) = $G \cdot \rho$ has units of S/cm². The definition of ρ was added in the revised manuscript. The ρ value adopted for $n=3$ and $n=4$ graphyne membranes are $2.39 \times 10^{18}/\text{cm}^2$ and $1.64 \times 10^{18}/\text{cm}^2$, respectively.

Comment#4:

Paper needs editing for English style and usage. For instance,

a. In the abstract line 10, “Despite of its remarkable features resulted from the use of liquid fuel” sounds wrong. It should be ‘resulting from’ and ‘Despite its...’.

b. On page 2 line 35, “Along with the emergence of two-dimensional (2D) materials since the first discovery of graphene [9-11], 2D materials with artificially created pores become a new choice of membrane..”

c. On page 3 line 51, “Graphyne is a family of 2D materials possess uniform triangular pores with controllable pore size...”

d. On page 11, line 212, it is ‘Nernst-Einstein’ and not ‘Nernest-Einstein’.

I am only listing a few instances. Please have the paper proof-read for English usage.

Response:

We thank the reviewer for this comment. We have proof-read the paper and revised the manuscript accordingly.

We thank the reviewer for the valuable suggestions, which are helpful for improving the quality of this manuscript.

References

- [R1] Videa, M., Xu, W., Geil, B., Marzke, R. & Angell, C. A. High Li⁺ self-diffusivity and transport number in novel electrolyte solutions. *J. Electrochem. Soc.* **148**, A1352-A1356 (2001)
- [R2] MacFarlane, D. R., Forsyth, M., Izgorodina, E. I., Abbott, A. P., Annat, G. & Fraser, K. On the concept of ionicity in ionic liquids. *Phys. Chem. Chem. Phys.* **11**, 4962-4967 (2009)
- [R3] Sangoro, J. R., Serghei, A., Naumov, S., Galvosas, P., Kärger, J., Wespe, C., Bordusa, F. & Kremer, F. Charge transport and mass transport in imidazolium-based ionic liquids. *Phys. Rev. E.* **77**, 051202 (2008)

REVIEWERS' COMMENTS:

Reviewer #1 (Remarks to the Author):

I consider the paper publishable since all indicated minor issues have been properly addressed.

Reviewer #2 (Remarks to the Author):

The authors have addressed all concerns I had with the manuscript, the next steps will be to test their computational findings experimentally which is beyond the scope of the present work. I am supportive of publication with minimal change.

Reviewer #3 (Remarks to the Author):

The authors have performed additional calculations and provided explanations for the concerns raised in the previous review. Removing the constraints have reduced the barrier for proton transfer through (n=2) graphyne but is still high to be overcome at room temperature. Therefore, the findings in the manuscript remain unchanged.

The authors have also addressed the concerns in English style and usage. I recommend the manuscript for publication.